# Symbiotic Microorganisms and Their Different Association Types in Aquatic and Semiaquatic Bugs

Yu Men,[a] Zi-wen Yang,[a] Jiu-yang Luo,[a] Ping-ping Chen,[b] Felipe Ferraz Figueiredo Moreira,[c] Zhi-hui Liu,[a] Jia-dong Yin,[a] Bao-jun Xie,[a] Yan-hui Wang,[a] Qiang Xie[a]

[a]School of Life Sciences, State Key Laboratory of Biocontrol, Sun Yat-sen University, Guangzhou, Guangdong, China
[b]Netherlands Centre of Biodiversity Naturalis, Leiden, Netherlands
[c]Laboratório de Biodiversidade Entomológica, Instituto Oswaldo Cruz, Fundação Oswaldo Cruz, Rio de Janeiro, Brazil

**ABSTRACT** True bugs (Hemiptera, suborder Heteroptera) constitute the largest suborder of nonholometabolous insects and occupy a wide range of habitats various from terrestrial to semiaquatic to aquatic niches. The transition and occupation of these diverse habitats impose various challenges to true bugs, including access to oxygen for the aquatic species and plant defense for the terrestrial phytophagans. Although numerous studies have demonstrated that microorganisms can provide multiple benefits to terrestrial host insects, a systematic study with comprehensive higher taxa sampling that represents aquatic and semiaquatic habitats is still lacking. To explore the role of symbiotic microorganisms in true bug adaptations, 204 samples belonging to all seven infraorders of Heteroptera were investigated, representing approximately 85% of its superfamilies and almost all known habitats. The symbiotic microbial communities of these insects were analyzed based on the full-length amplicons of the bacterial 16S rRNA gene and fungal ITS region. Bacterial communities varied among hosts inhabiting terrestrial, semiaquatic, and aquatic habitats, while fungal communities were more related to the geographical distribution of the hosts. Interestingly, co-occurrence networks showed that species inhabiting similar habitats shared symbiotic microorganism association types. Moreover, functional prediction analyses showed that the symbiotic bacterial community of aquatic species displayed richer amino acid and lipid metabolism pathways, while plant-feeding true bugs benefited more from the symbiont-provided xenobiotics biodegradation pathway. These results deepened the recognition that symbiotic microorganisms were likely to help heteropterans occupy diverse ecological habitats and provided a reference framework for further studies on how microorganisms affect host insects living in various habitats.

**IMPORTANCE** Symbiotic bacteria and fungi generally colonize insects and provide various benefits for hosts. Although numerous studies have investigated symbionts in terrestrial plant-feeding insects, explorations of symbiotic bacterial and fungal communities in aquatic and semiaquatic insects are rare. In this study, the symbiotic microorganisms of 204 aquatic, semiaquatic, and terrestrial true bugs were explored. This comprehensive taxon sampling covers ~85% of the superfamilies of true bugs and most insect habitats. Analyses of the diversity of symbionts demonstrated that the symbiotic microbial diversities of true bugs were mainly affected by host habitats. Co-occurrence networks showed that true bugs inhabiting similar habitats shared symbiotic microbial association types. These correlations between symbionts and hosts together with the functions of bacterial communities indicated that symbiotic microbial communities may help true bugs adapt to (semi)aquatic habitats.

**KEYWORDS** aquatic/semiaquatic bugs, co-occurrence network, diverse habitats, Heteroptera, symbiotic microorganism

Address correspondence to Yan-hui Wang, wangyanh3@mail.sysu.edu.cn.

The authors declare no conflict of interest.

True bugs (Hemiptera, suborder Heteroptera) constitute an ecologically unique group of insects comprising 91 families and more than 45,000 described species (1). It is the only major group successfully utilizing almost all habitats occupied by insects, except for mining into plant tissues or internally parasitizing animals. In addition to living in numerous habitats, they can also utilize a wide range of food sources (2). The Heteroptera are divided into seven infraorders, i.e., Dipsocoromorpha, Enicocephalomorpha, Gerromorpha, Nepomorpha, Leptopodomorpha, Cimicomorpha, and Pentatomomorpha (3). The first five infraorders are intimate with aquatic environments, while the last two infraorders are terrestrial species.

Compared with terrestrial organisms, aquatic species face entirely different challenges due to the features of water, i.e., high viscosity, solvent properties, and low concentration of oxygen. Their survival, health, growth, reproduction, and abundance can be affected by metal pollution (4, 5), contaminants (6, 7), harmful algal blooms (6), and salinity stress (8). More importantly, aquatic species are challenged by the most profound and extensive factor, hypoxia (9, 10). For phytophagous true bugs, the imbalanced amino acid profiles and toxic chemicals in plant tissues constitute different challenges (11, 12). Whether different survival challenges are accompanied by different compositions of symbiotic microorganisms remains unknown.

Previous studies of symbiotic microorganisms mainly focused on terrestrial-plant-feeding insects. The bacterial communities of honeybees, fruit flies, beetles, silkworms, whiteflies, and aphids depend on diet, habitats, domestication, and environmental factors (13–16), help hosts with nutrition, digestion, and detoxification, and facilitate changing of host plants (17–20). In contrast, studies on the symbiotic microorganisms of aquatic and carnivorous insects are still limited. True bugs are a group of ideal insects with which to explore this scientific issue. Species of the five infraorders mentioned above are carnivorous and rely on aquatic environments to various degrees. Most species of Dipsocoromorpha and Enicocephalomorpha are ground dwellers, especially in humid areas. Except for lucky encounters with swarming individuals of Enicocephalomorpha, insects of these two infraorders are difficult to effectively collect by regular means, such as net sweeping or light trapping (21). Species of Gerromorpha and Leptopodomorpha mainly live on the water surface and in riparian habitats and are, thus, called semiaquatic bugs and shore bugs, respectively (22). Those belonging to Nepomorpha are commonly regarded as truly aquatic bugs that live submerged, except for the superfamily Ochteroidea, which underwent a reversal to living in riparian habitats (23). Comparatively, species of Cimicomorpha and Pentatomomorpha are terrestrial and mostly found on vegetation. The common ancestor of the true bugs is generally recognized as terrestrial/ground living (24), and the transitions from terrestrial to semiaquatic and aquatic habitats occurred independently from the late Permian to the early Triassic (269 to 246 Ma) (25–27). Regarding feeding habits, all typical plant-feeding species belong to Pentatomomorpha and the superfamily Muroidea in Cimicomorpha, while the remaining species are mainly predatory (27). In true bugs, the symbiotic bacteria of terrestrial pentatomomorphans have been demonstrated to provide multiple benefits for hosts, including digestion and nutrition (19), as well as survival, development, reproduction, and adaptation (28–33). The symbiotic bacteria of certain cimicomorphans in the superfamilies Reduvioidea and Miroidea were influenced by ontogeny, species identity, and the environment (34) and were linked to nutrition (35) and survival (36). In contrast, until now, studies on symbiotic bacterial and fungal communities of the other five infraorders of Heteroptera inhabiting semiaquatic and aquatic habitats have been rare. Only two works mentioned the symbiotic bacteria of six gerromorphan species, which suggested that the bacterial composition was influenced by the host (37) and salinity (38). At the same time, there is no study on the symbiotic fungal community of true bugs, which makes it difficult to comprehensively determine the symbiotic microbial communities of aquatic and semiaquatic bugs, the overall pattern of symbiotic microorganisms of all true bugs, and further the microbial roles in their adaptation to diverse habitats.

**TABLE 1** Number of samples belonging to each infraorder and superfamily

| Infraorder | Superfamily | Family | Fungal sample | Bacterial sample |
|---|---|---|---|---|
| Dipsocoromorpha[a] | | 3 | 2 | 7 |
| Enicocephalomorpha[a] | | 2 | 1 | 2 |
| Gerromorpha[a] | Mesoveloidea | 1 | 2 | 5 |
| Gerromorpha[a] | Hebroidea | 1 | 1 | 6 |
| Gerromorpha[a] | Hydrometroidea | 1 | 1 | 4 |
| Gerromorpha[a] | Gerroidea | 3 | 10 | 41 |
| Nepomorpha[a] | Corixoidea | 2 | 4 | 18 |
| Nepomorpha[a] | Nepoidea | 2 | 2 | 10 |
| Nepomorpha[a] | Ochteroidea | 2 | 6 | 7 |
| Nepomorpha[a] | Notonectoidea | 3 | 1 | 16 |
| Nepomorpha[a] | Naucoroidea | 3 | 2 | 7 |
| Leptopodomorpha[a] | Leptopodoidea | 2 | 4 | 5 |
| Leptopodomorpha[a] | Saldoidea | 1 | 5 | 11 |
| Cimicomorpha | Reduvioidea | 1 | 1 | 5 |
| Cimicomorpha | Miroidea | 2 | 6 | 12 |
| Cimicomorpha | Naboidea | 1 | 1 | 3 |
| Cimicomorpha | Cimicoidea | 1 | 3 | 4 |
| Pentatomomorpha | Aradoidea | 1 | 2 | 2 |
| Pentatomomorpha | Pentatomoidea | 9 | 8 | 17 |
| Pentatomomorpha | Pyrrhocoroidea | 2 | 5 | 8 |
| Pentatomomorpha | Coreoidea | 3 | 1 | 7 |
| Pentatomomorpha | Lygaeoidea | 6 | 3 | 7 |
| | Total | 52 | 71 | 204 |

[a]Samples in these infraorders were newly collected and sequenced in this study.

In this study, analyses of symbiotic microbial diversity across all seven infraorders of Heteroptera with comprehensive taxon sampling were accomplished. In total, 204 samples representing almost all known habitats of true bugs were sampled. Full-length bacterial 16S rRNA genes and fungal internal transcribed spacer (ITS) regions were amplified and then sequenced with the PacBio platform. Analyses of microbial composition, diversity, co-occurrence network, and function prediction were performed to investigate the role of symbiotic bacterial and fungal communities in host adaptations to aquatic and semiaquatic habitats.

## RESULTS

**Composition of bacterial and fungal communities.** The compositions of symbiotic bacteria of 204 samples belonging to seven infraorders, 20 superfamilies, and 52 families of Heteroptera were surveyed (Table 1; Table S1 in Supplemental File 1). After quality control, a total of 1,019,323 sequences and 3,063 amplicon sequence variants (ASVs) were obtained (Table S2 in Supplemental File 1). The dominant bacterial phyla were *Proteobacteria* (71.41%) and *Firmicutes* (14.24%) (Fig. 1A). The bacterial communities of Nepomorpha (aquatic) and Gerromorpha (semiaquatic) contained several unique bacterial phyla compared with those of the other five infraorders of true bugs. The phylum *Deferribacterota* was mainly present in Nepomorpha, while the phyla *Desulfobacterota* and *Bacteroidota* were mainly present in Gerromorpha and Nepomorpha (Fig. 1A). In addition, the orders *Oscillospirales* (*Firmicutes*: *Clostridia*) and *Erysipelotrichales* (*Firmicutes*:*Bacilli*) were mainly present in Nepomorpha, while the order *Rhodospirillales* (*Proteobacteria*: *Alphaproteobacteria*) was mainly present in Nepomorpha and Gerromorpha (Fig. S1A in Supplemental File 1).

Fungal data from a total of 71 samples belonging to seven infraorders, 20 superfamilies, and 38 families of Heteroptera were obtained (Table 1; Table S1 in Supplemental File 1). After quality control, 387,572 sequences and 1,173 ASVs were obtained (Table S2 in Supplemental File 1). The dominant fungal phyla were *Ascomycota* (51.53%), *Basidiomycota* (26.01%), and *Chytridiomycota* (11.66%). In Cimicomorpha (terrestrial), the phylum *Ascomycota* dominated in all samples. In the other six infraorders of true bugs, the dominant fungal phyla varied among samples (Fig. 1B). Correspondingly, no fungal order dominated in any of the seven infraorders (Fig. S1B in Supplemental File 1).

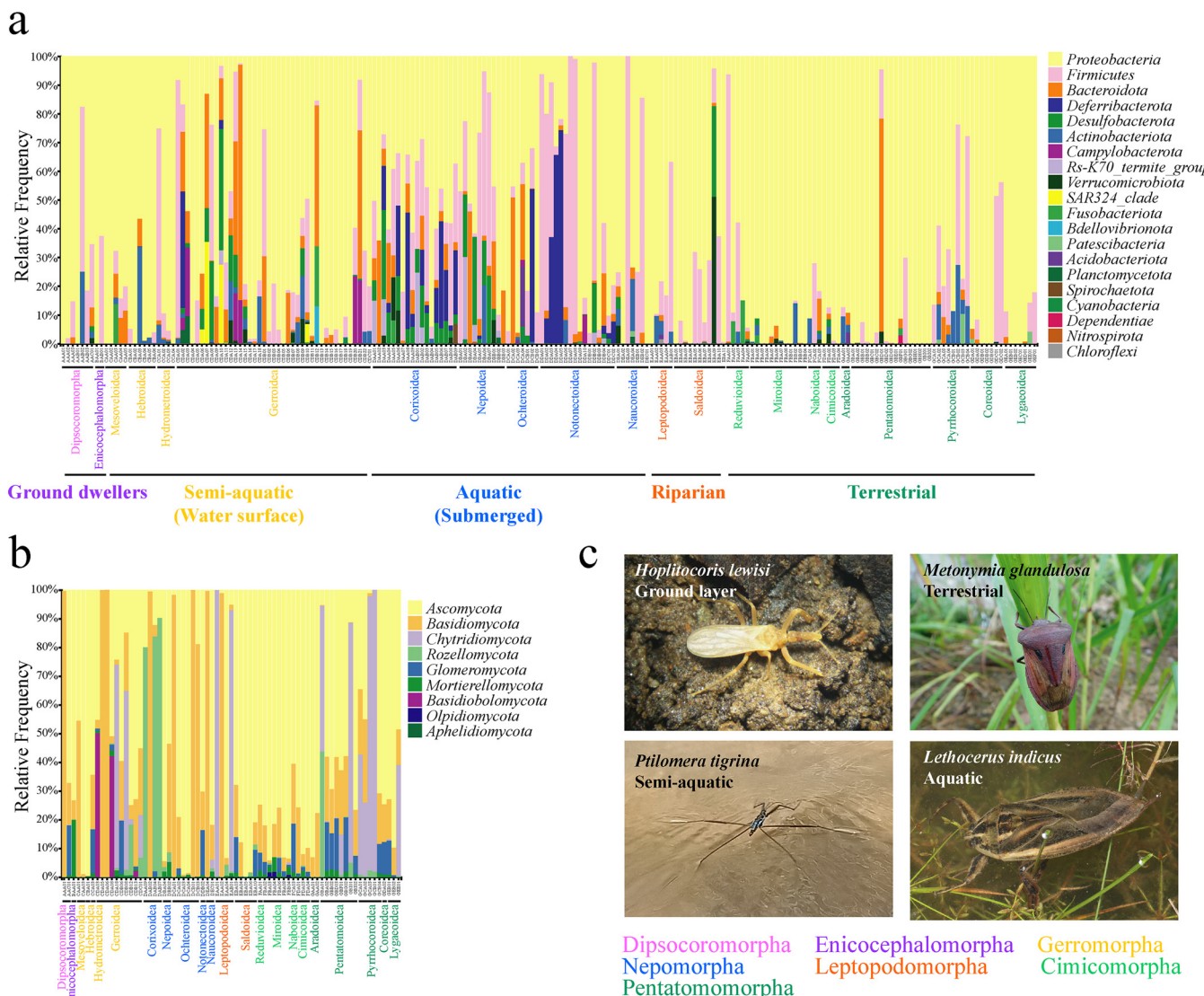

**FIG 1** Composition of symbiotic bacterial and fungal communities in true bugs. Relative abundance plots of bacterial (A) and fungal (B) phyla of 204 and 71 samples are shown, respectively. The host infraorders or superfamilies are shown at the bottom of each plot and are colored according to the infraorders or to the infraorders to which they belong. (C) Representative images of true bugs inhabiting four typical habitats.

**Factors influencing bacterial and fungal communities.** Alpha diversity results showed that symbiotic bacterial communities varied among hosts inhabiting different habitats and between different environments (i.e., geographical sites, altitudes, and temperature zones). In detail, four alpha diversity indices (Faith's phylogenetic diversity [PD], observed features, Pielou's evenness, and Shannon entropy indices) were calculated. Then, three statistical tests (phylogenetic analysis of variance [ANOVA] test, Fisher's least significant distance [LSD] *post hoc* test, and Kruskal-Wallis test) were performed to evaluate differences between each pair of groups. The Faith's PD index of bacterial communities in aquatic bugs (Nepomorpha) was the highest and significantly different from that of terrestrial bugs (Cimicomorpha and Pentatomomorpha) (LSD *post hoc* test and Kruskal-Wallis test; adjusted $P < 0.05$) (Fig. 2A; Table 2; Table S3 in Supplemental File 1). While, the observed features, Pielou's evenness, and Shannon entropy indices showed no significant difference among aquatic and terrestrial bugs (Fig. S2A, C, and E in Supplemental File 1; Table S3 in Supplemental File 1). The different results manifested by various alpha diversity indices and statistical tests may be attributed to the habitats of true bugs, which mainly have effects on the composition and phylogenetic diversity of symbiotic bacteria. For the environmental factors, except

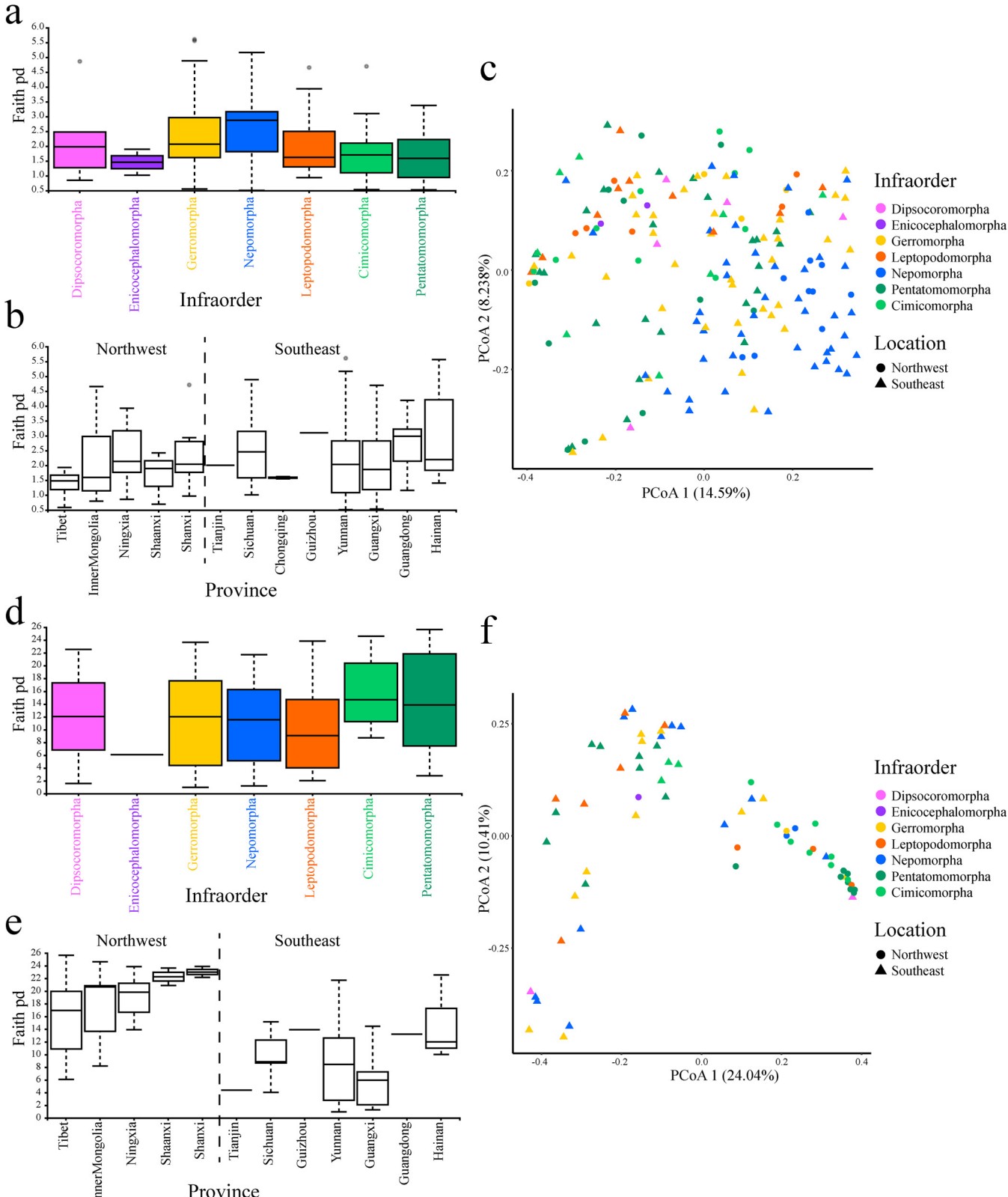

**FIG 2** The alpha and beta diversity of symbiotic microbial communities. The samples are grouped according to geographic position and infraorder. (A to C) are the diversity results of bacterial communities. Faith's PD indices of symbiotic bacterial communities are mainly affected by hosts and habitats (A) but are not correlated with geographical positions (B). (C) The PCoA plot of bacterial communities based on unweighted UniFrac distance. (D to F) are the diversity results of fungal communities. Faith's PD index of symbiotic fungal communities is mainly affected by geographical positions (E) but is not correlated with hosts (D). (F) The PCoA plot of symbiotic fungal communities based on unweighted UniFrac distance.

**TABLE 2** Medians of alpha diversity indices in each group

| Characteristic | Bacteria | | | | Fungi | | | |
|---|---|---|---|---|---|---|---|---|
| | Faith's PD | Observed features | Pielou's evenness | Shannon entropy | Faith's PD | Observed features | Pielou's evenness | Shannon entropy |
| **Infraorder** | | | | | | | | |
| Dipsocoromorpha | 1.98 | 20.00 | 0.71 | 2.95 | 12.06 | 71.00 | 0.42 | 2.91 |
| Enicocephalomorpha | 1.46 | 15.00 | 0.81 | 2.92 | 6.08 | 37.00 | 0.82 | 4.27 |
| Gerromorpha | 2.07 | 15.00 | 0.72 | 2.67 | 12.01 | 61.00 | 0.81 | 5.00 |
| Nepomorpha | 2.88 | 20.00 | 0.74 | 3.22 | 11.53 | 45.50 | 0.43 | 2.23 |
| Leptopodomorpha | 1.63 | 17.00 | 0.70 | 2.96 | 9.07 | 29.00 | 0.69 | 3.29 |
| Cimicomorpha | 1.71 | 12.00 | 0.70 | 2.11 | 14.69 | 65.00 | 0.68 | 3.67 |
| Pentatomomorpha | 1.59 | 23.00 | 0.92 | 4.11 | 13.85 | 74.50 | 0.83 | 5.07 |
| **Province** | | | | | | | | |
| Tibet | 1.48 | 8.00 | 0.64 | 1.92 | 16.94 | 73.50 | 0.74 | 4.51 |
| InnerMongolia | 1.60 | 9.00 | 0.64 | 2.03 | 20.68 | 87.00 | 0.70 | 4.52 |
| Ningxia | 2.14 | 16.00 | 0.52 | 1.78 | 19.84 | 95.00 | 0.77 | 5.10 |
| Shaanxi | 1.90 | 15.00 | 0.62 | 2.40 | 22.27 | 141.00 | 0.73 | 5.25 |
| Shanxi | 2.04 | 13.00 | 0.41 | 1.69 | 23.01 | 139.50 | 0.86 | 6.10 |
| Tianjin | 2.01 | 27.00 | 0.91 | 4.32 | 4.40 | 19.00 | 0.64 | 2.74 |
| Sichuan | 2.46 | 18.50 | 0.67 | 2.78 | 8.88 | 29.00 | 0.35 | 1.75 |
| Chongqing | 1.57 | 8.50 | 0.65 | 2.00 | | | | |
| Guizhou | 3.10 | 15.00 | 0.32 | 1.27 | 13.92 | 69.00 | 0.74 | 4.55 |
| Yunnan | 2.03 | 23.00 | 0.89 | 3.82 | 8.45 | 30.00 | 0.79 | 4.10 |
| Guangxi | 1.86 | 26.00 | 0.96 | 4.37 | 5.96 | 23.00 | 0.58 | 2.63 |
| Guangdong | 2.98 | 19.00 | 0.78 | 3.35 | 13.23 | 63.00 | 0.79 | 4.70 |
| Hainan | 2.20 | 28.00 | 0.85 | 3.62 | 12.01 | 61.00 | 0.81 | 5.24 |
| **Altitude** | | | | | | | | |
| Low altitude | 2.07 | 19.00 | 0.74 | 3.04 | 10.24 | 49.00 | 0.79 | 4.55 |
| Middle altitude | 1.93 | 20.50 | 0.77 | 3.18 | 12.92 | 64.00 | 0.71 | 4.17 |
| Sub-high altitude | 1.27 | 6.00 | 0.59 | 1.45 | 16.37 | 65.00 | 0.71 | 4.27 |
| High altitude | 1.24 | 16.00 | 0.92 | 3.68 | | | | |
| **Temperature zone** | | | | | | | | |
| Plateau temperate zone | 1.38 | 6.00 | 0.67 | 1.34 | 9.07 | 37.00 | 0.50 | 3.04 |
| Temperate zone | 2.03 | 12.00 | 0.63 | 2.01 | 20.11 | 87.00 | 0.73 | 4.52 |
| Warm temperate zone | 1.83 | 14.00 | 0.56 | 1.96 | 22.17 | 125.00 | 0.81 | 5.42 |
| North subtropical zone | 1.90 | 15.00 | 0.62 | 2.40 | 22.27 | 141.00 | 0.73 | 5.25 |
| Mid-subtropical zone | 1.91 | 17.50 | 0.69 | 2.50 | 11.26 | 50.50 | 0.70 | 3.92 |
| South subtropical zone | 2.26 | 24.00 | 0.93 | 4.11 | 8.45 | 27.00 | 0.79 | 3.29 |
| Marginal tropical zone | 2.00 | 25.00 | 0.86 | 3.52 | 6.06 | 29.00 | 0.79 | 4.16 |
| Tropical zone | 2.13 | 23.00 | 0.95 | 4.19 | 11.02 | 55.00 | 0.85 | 4.90 |

Faith's PD index, the remaining three alpha diversity indices of samples collected from the southeast were significantly higher than that of samples collected from northwest China (Fig. 2B; Table 2; Fig. S2B, D, and F in Supplemental File 1; Table S3 in Supplemental File 1). The bacterial communities of samples collected from low and middle altitudes showed significantly higher observed features, Pielou's evenness, and Shannon entropy indices than that of samples collected from sub-high altitude (adjusted $P < 0.05$) (Fig. S3A, C, E, and G in Supplemental File 1). In addition, the alpha diversity indices of samples collected from warmer areas were higher than those of samples collected from colder areas (Fig. S3B, D, F, and H in Supplemental File 1).

The principal coordinate analysis (PCoA) plots based on unweighted and weighted UniFrac distances showed significantly different bacterial compositions between terrestrial and aquatic true bugs. In terrestrial true bugs, samples of Cimicomorpha had similar bacterial communities to those of Pentatomomorpha. While aquatic Nepomorpha presented completely different bacterial compositions. Comparatively, the bacterial communities of species in Gerromorpha and Leptopodomorpha were indistinguishable from those of terrestrial or aquatic bugs according to the PCoA plot (Fig. 2C; Fig. S4A in

Supplemental File 1). These results were supported by the pairwise permutational multivariate ANOVA (PERMANOVA) tests with $P$ and $q$ values $< 0.01$, especially the bacterial community of Nepomorpha, which was significantly different from that of other infraorders (Table S4 in Supplemental File 1).

To further assess the relationships between hosts and symbiotic bacteria, the Mantel test, and Procrustes analysis were performed to evaluate the correlations of unweighted UniFrac distance with genetic and geographical distances. The above results further verified that the bacterial communities were significantly correlated with the host and slightly correlated with geographical distance (Fig. S5 in Supplemental File 1).

In contrast to the pattern of the bacterial community, the symbiotic fungal community was related to the environment. Except for Pielou's evenness index, which showed that the evenness of terrestrial Pentatomomorpha was significantly higher than that of aquatic Nepomorpha (LSD *post hoc* test and Kruskal-Wallis test; adjusted $P < 0.05$), the remaining alpha diversity indices of terrestrial bugs (Cimicomorpha and Pentatomomorpha), and the bugs collected at higher altitudes were slightly higher than those of aquatic and semiaquatic bugs and the bugs collected at lower altitudes (phylogenetic ANOVA analysis, LSD *post hoc* test, and Kruskal-Wallis test; adjusted $P > 0.05$) (Fig. 2D; Table 2; Fig. S6A, C, and E in Supplemental File 1; Fig. S7A, C, E, and G in Supplemental File 1; Table S5 in Supplemental File 1). The true bugs collected from northwestern China and colder areas exhibited significantly higher alpha diversity indices than those collected from southeastern China and warmer areas (phylogenetic ANOVA analysis and Kruskal-Wallis test, adjusted $P < 0.01$) (Fig. 2E; Fig. S6B, D, and F in Supplemental File 1; Fig. S7B, D, F, and H in Supplemental File 1; Table S5 in Supplemental File 1). Regarding the beta diversity of fungal communities, the samples collected from southeastern and northwestern China separated from each other according to the PCoA plot (Fig. 2F; Fig. S4B in Supplemental File 1) and PERMANOVA tests ($P$ and $q$ values $< 0.01$; Table S4 in Supplemental File 1). The Mantel test and Procrustes analysis provided further evidence that fungal communities were significantly correlated with geographical distance and were slightly correlated with hosts (Fig. S5 in Supplemental File 1).

**Pattern of abundant symbionts according to the phylogeny of host insects.** To further investigate the relationships between true bugs and their symbiotic microorganisms, the phylogeny of true bugs was inferred using nearly complete cytochrome C oxidase subunit I (COI), COII, 18S, and 28S rRNA genes. The monophyly for most nodes was well supported, with bootstrap values greater than 90% (Fig. 3; Fig. S8 and S9 in Supplemental File 1). The relative abundances of the top 14 bacterial and 14 fungal orders corresponding to the host phylogeny were shown (Fig. 3). Note that true bugs with similar habitats had similar symbiotic bacterial compositions and abundances. Most host superfamilies were dominated by the orders *Enterobacterales* (*Proteobacteria*: *Gammaproteobacteria*), *Rickettsiales* (*Proteobacteria*: *Alphaproteobacteria*), and *Burkholderiales* (*Proteobacteria*: *Gammaproteobacteria*). In aquatic Nepomorpha, however, all superfamilies except Ochteroidea had no quantitatively dominant bacteria. Although the bacterial communities in Ochteroidea were dominated by the order *Enterobacterales*, the abundances of the orders *Bacteroidales* (*Bacteroidota*: *Bacteroidia*) and *Deferribacterales* (*Deferribacterota*: *Deferribacteres*) were still similar to those of the other species of Nepomorpha. For symbiotic fungi, an unidentified order in *Rozellomycota* was mainly present in the superfamily Corixoidea of Nepomorpha. In addition, another unidentified order in *Chytridiomycota* was mainly present in Gerromorpha, Leptopodomorpha, and Pentatomomorpha. The relative abundance of the order *Filobasidiales* (*Basidiomycota*: *Tremellomycetes*) was lower in terrestrial bugs than in aquatic and semiaquatic bugs (Fig. 3).

At the genus level, the losses and acquisitions of symbionts among host superfamilies were also provided (Fig. S9 in Supplemental File 1). For bacterial communities, the genus *Wolbachia* (*Proteobacteria*: *Alphaproteobacteria*) was present in all seven infraorders. An uncultured genus in the order *Enterobacterales* and the genus *Rickettsiella* (*Proteobacteria*: *Gammaproteobacteria*) were absent in Leptopodomorpha, Cimicomorpha, and Pentatomomorpha. In other words, these bacteria were only present in the remaining infraorders, in which most bugs live in humid and (semi)aquatic areas. In addition, there was no *Burkholderia* (*Proteobacteria*:

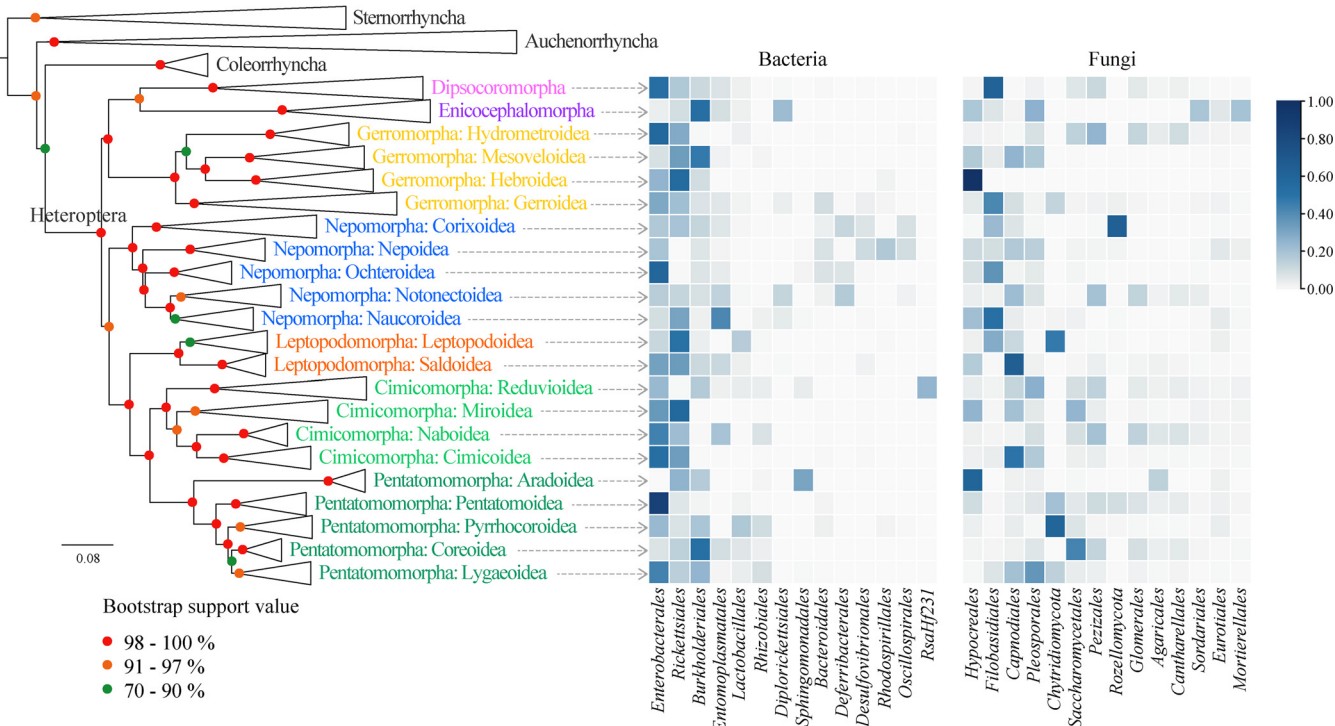

**FIG 3** Phylogenetic relationships of true bugs and the corresponding heatmap of the abundance of symbiotic microbial communities at the order level. Branches were collapsed, and only the infraorder and superfamily names were provided. Internal tree nodes are labeled with colored dots. The heatmap on the left represents the 14 most abundant bacterial orders. The heatmap on the right represents the 14 most abundant fungal orders. For the unidentified orders, their lowest-level classifications are given instead.

*Gammaproteobacteria*) in aquatic Nepomorpha. The genera *Pantoea*, *Pectobacterium*, and *Caballeronia* (*Proteobacteria*: *Gammaproteobacteria*) were mainly present in terrestrial bugs. In fungal communities, the genus *Naganishia* (*Basidiomycota*: *Tremellomycetes*) was absent in terrestrial bugs (Cimicomorpha and Pentatomomorpha). An unidentified genus in the phylum *Chytridiomycota* was absent in terrestrial Cimicomorpha as well.

To determine whether significantly different bacteria and fungi existed in different infraorders, LEfSe analyses were performed at both the genus level (Fig. S10A and C in Supplemental File 1) and order level (Fig. S10B in Supplemental File 1). In each infraorder, several specialized bacterial genera and orders showed significant increases in terms of abundance (Fig. S10A and B in Supplemental File 1), such as the genus *Caballeronia* in terrestrial Pentatomomorpha, the genera *Wolbachia* and *Rickettsia* in terrestrial Cimicomorpha, and the genera *Mucispirillum* (*Deferribacterota*: *Deferribacteres*), *Desulfovibrio* (*Desulfobacterota*: *Desulfovibrionia*), and *Dysgonomonas* (*Bacteroidota*: *Bacteroidia*) in aquatic Nepomorpha. Meanwhile, only the symbiotic fungal genera of Enicocephalomorpha and Cimicomorpha showed significant increases in terms of abundance. Comparatively, the results showed that no significantly different fungal order was found.

**Differences of symbiotic microorganism association types in host insects.** To explore the relationships between symbiotic bacterial and fungal communities, the Mantel test and Procrustes analysis were performed based on unweighted UniFrac distances. The results of the Mantel test and Procrustes analysis showed significant correlations (*P* = 0.039) and no statistical relevance (*P* = 0.079) between bacterial and fungal communities, respectively (Fig. S11 in Supplemental File 1). The difference reflected different levels of associations among various symbionts.

More detailed associations were further analyzed by co-occurrence networks. In each infraorder, three co-occurrence networks were evaluated, respectively, i.e., the co-occurrence network of the 120 most abundant bacterial ASVs, the co-occurrence network of the 120 most abundant fungal ASVs, and the co-occurrence network of 240 bacterial and fungal ASVs overall (Fig. 4; Table 3; Fig. S12 in Supplemental File 1). The

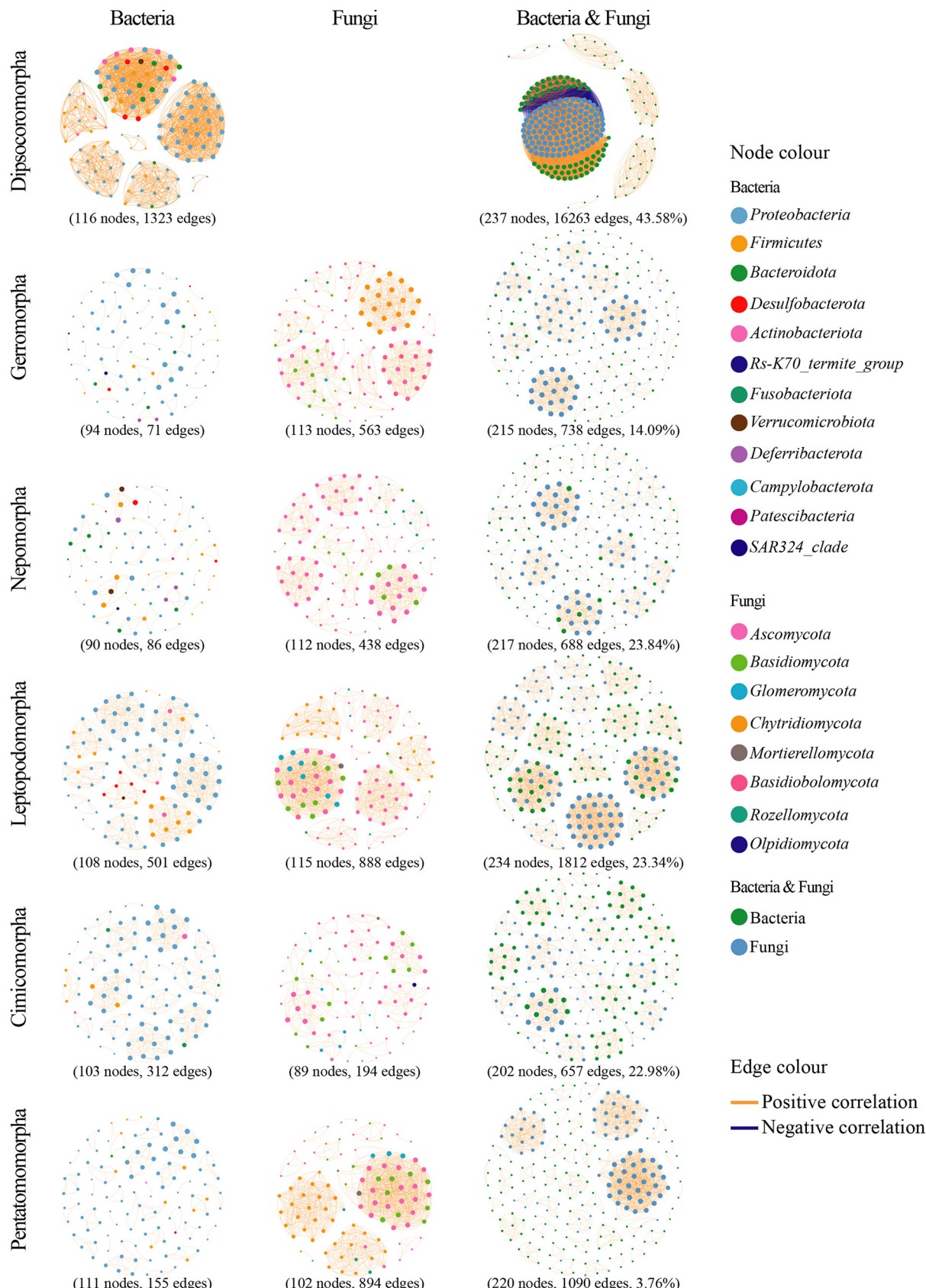

**FIG 4** Co-occurrence networks of the symbiotic microbial community. For each infraorder, the co-occurrence network of bacterial communities (left column), the co-occurrence network of fungal communities (middle column), and the co-occurrence network of bacterial

three networks of Enicocephalomorpha and the fungal network of Dipsocoromorpha were not implemented due to insufficient samples (<4 samples). The symbiotic bacterial communities of Dipsocoromorpha exhibited the largest potential interaction network (116 nodes and 1 323 edges), followed successively by those of Leptopodomorpha (108 nodes and 501 edges) and Cimicomorpha (103 nodes and 312 edges). Several distinct clustering groups were found in the networks of these three infraorders. The networks for the bacterial communities of Nepomorpha (90 nodes and 86 edges) and Gerromorpha (94 nodes and 71 edges) were relatively simpler, with the coordinated ASVs belonging to more phyla.

Compared with bacteria, the networks of fungal communities were more complex (Fig. 4). The symbiotic fungi of Leptopodomorpha (115 nodes and 888 edges) and Pentatomomorpha (102 nodes and 894 edges) presented relatively complex interaction networks, while that of Cimicomorpha (89 nodes and 194 edges) was the simplest. Although approximate numbers of nodes were shown in the fungal networks of Gerromorpha, Nepomorpha, Leptopodomorpha, and Pentatomomorpha, the numbers of edges in Nepomorpha and Gerromorpha were approximately half of those in Pentatomomorpha and Leptopodomorpha. All networks showed several groups with distinct clustering ASVs, except that of Cimicomorpha.

Moreover, the overall interactions among the top 240 bacterial and fungal ASVs were evaluated together (Fig. 4). The percentages of edges representing interactions between bacteria and fungi varied from 3.76% in Pentatomomorpha to 43.58% in Dipsocoromorpha. Notably, in the network of Dipsocoromorpha, a group of bacterial ASVs showed negative correlations with a group of fungal ASVs. In the group of bacterial ASVs, 77% belonged to the family *Budviciaceae* (*Proteobacteria*: *Gammaproteobacteria*). The two negatively correlated groups of bacterial and fungal symbionts may occupy similar niches in hosts. Additionally, almost all ASVs in the remaining co-occurrence networks were positively correlated, suggesting potential cooperation as well as positive associations among bacterial and fungal communities of true bugs.

**Functional predictions of symbiotic bacterial and fungal communities.** Considering the diversity of the symbiotic communities and the potential coordination among them, the functions of the symbiotic bacterial and fungal communities were predicted and compared. The results showed that the symbiotic bacterial communities may provide multiple benefits for hosts, including but not limited to amino acid metabolism, lipid metabolism, metabolism of cofactors and vitamins, xenobiotics biodegradation and metabolism, and the immune system (Fig. 5; Fig. S13 to S19 in Supplemental File 1). For each infraorder, the symbiotic bacterial functions were compared with those of the remaining true bugs. For the category of amino acid metabolism, both the metabolic pathways of alanine, aspartate, and glutamate and the biosynthesis pathways of phenylalanine, tyrosine, and tryptophan accounted for a significantly high proportion in the symbiotic bacterial communities of aquatic bugs. While the metabolic pathways of arginine, proline, phenylalanine, tryptophan, and tyrosine, and the degradation pathway of lysine accounted for significantly low proportions in the symbiotic bacterial communities of aquatic bugs. In addition, the pathways belonging to lipid metabolism were significantly different between aquatic bugs and all other heteropteran infraorders. Furthermore, the nucleotide-binding oligomerization domain (NOD)-like receptor signaling pathway belonging to the immune system was significantly higher in aquatic bugs. Finally, the proportions of pathways belonging to xenobiotics biodegradation and metabolism were significantly higher in the symbiotic bacterial communities of terrestrial pentatomomorphans and were significantly lower in the symbiotic bacterial

**FIG 4** Legend (Continued)
and fungal communities (right column) are shown. No symbiotic network of Enicocephalomorpha or fungal network of Dipsocoromorpha was analyzed because of the small sample sizes. Only correlations with |r| > 0.8 and *P* < 0.0001 are shown. Each node represents an ASV. The nodes are colored according to phylum (left and middle columns) or kingdom (right column). The node size is proportional to the weighted degree of ASVs. Edge thickness is proportional to the weight of correlation.

**TABLE 3** Overall features of symbiotic microbial communities

| Infraorder | Dipsocoromorpha | Enicocephalomorpha | Gerromorpha | Nepomorpha | Leptopodomorpha | Cimicomorpha | Pentatomomorpha |
|---|---|---|---|---|---|---|---|
| **Nature** | | | | | | | |
| Major habitat | Ground dwellers | Ground dwellers | Water surface | Submerged | Riparian | Terrestrial | Terrestrial |
| Trivial name | Ground dwellers | Ground dwellers | Semiaquatic bugs | Truly aquatic bugs | Shore bugs | Terrestrial bugs | Terrestrial bugs |
| Major feeding habits | Predatory | Predatory | Predatory | Predatory | Predatory | Both | Phytophagous |
| **Diversity** | | | | | | | |
| No. of symbiotic bacterial phyla | 6 | 2 | 14 | 16 | 9 | 7 | 9 |
| Symbiotic bacterial phyla with relative abundance > 60% | Proteobacteria | Proteobacteria | Proteobacteria | None | Proteobacteria | Proteobacteria | Proteobacteria |
| No. of symbiotic fungal phyla | 5 | 3 | 8 | 6 | 7 | 7 | 6 |
| Symbiotic fungal phyla with relative abundance > 60% | Basidiomycota | None | None | None | Ascomycota | Ascomycota | None |
| **Co-occurrence network** | | | | | | | |
| No. of nodes + edges in bacterial network | Maximum | NA[b] | Minimum | Limited | Enriched | Enriched | Limited |
| No. of nodes + edges in fungal network | NA[b] | NA[b] | Limited | Limited | Maximum | Minimum | Enriched |
| No. of nodes + edges in bacterial and fungal network | Maximum | NA[b] | Limited | Limited | Enriched | Minimum | Enriched |
| Interactions between bacterial and fungal communities[a] | Maximum | NA[b] | Limited | Enriched | Enriched | Enriched | Minimum |
| **Functional prediction** | | | | | | | |
| More regulation in amino acid metabolism | No[c] | No[c] | No[c] | Yes | No[c] | No[c] | No[c] |
| More regulation in lipid metabolism | No[c] | No[c] | No[c] | Yes | No[c] | No[c] | No[c] |
| Xenobiotics biodegradation and metabolism | Limited | Limited | No[c] | Limited | Limited | No[c] | Enriched |

[a]The interactions between bacterial and fungal communities were evaluated according to the percentage of edges linking bacteria and fungi.
[b]Not applicable. The networks of Enicocephalomorpha and the fungal network of Dipsocoromorpha were not implemented due to insufficient samples (<4).
[c]No significant difference was shown between the specific infraorder and all other samples.

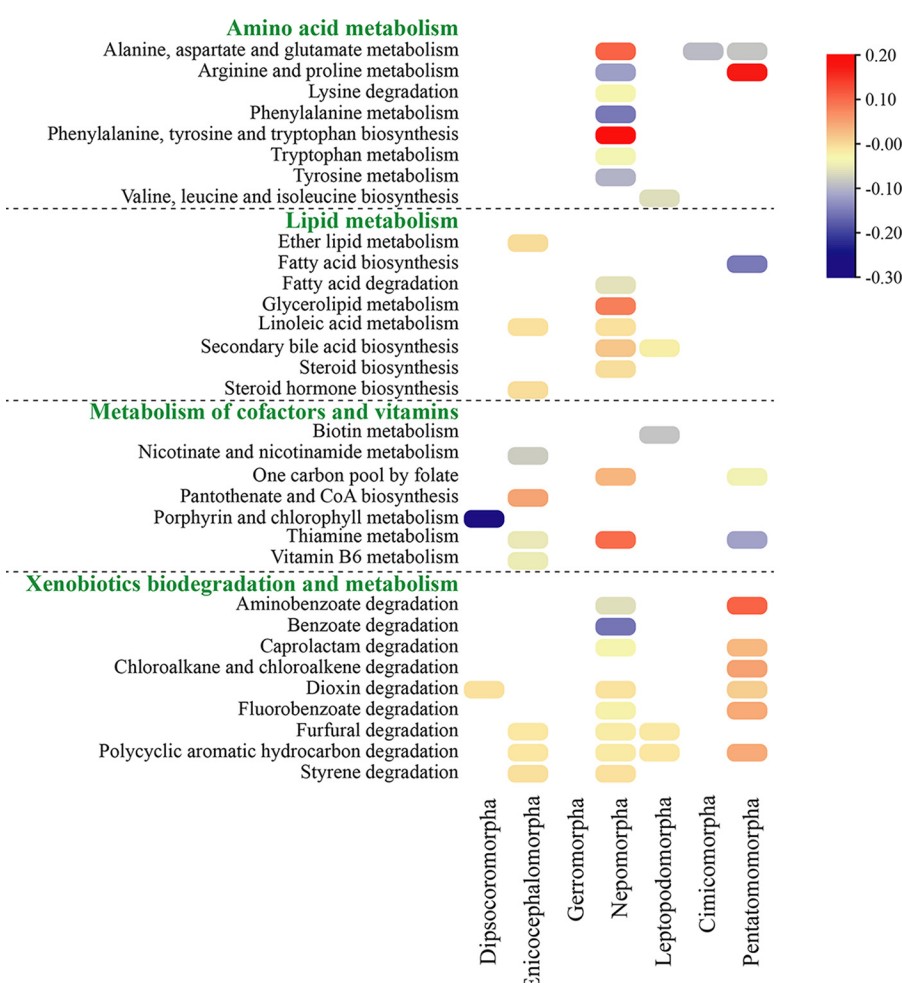

**FIG 5** Heatmap of functionally predicted pathways of symbiotic bacterial communities. For each infraorder, the proportions of pathways were compared with those of all other samples. The color represents the difference in mean proportions between a specific infraorder and the remaining samples. Only the pathways with $P < 0.001$ (Welch's $t$ test) are shown in the heatmap.

communities of the other infraorders.

Among the functional guilds of fungal communities, the relative abundances of undefined saprotroph (saprotroph), plant pathogen (pathotroph), and arbuscular mycorrhizal (symbiotroph) were the highest (Fig. S20 in Supplemental File 1). In addition, the symbiotic fungal guilds of each infraorder were compared with those of the remaining samples. Fungal communities of Enicocephalomorpha, Cimicomorpha, and Pentatomomorpha showed no significantly different functional guild. Fungal parasite (pathotroph), endophyte-plant pathogen (pathotroph-symbiotroph), ericoid mycorrhizal (pathotroph-symbiotroph), dung saprotroph-wood saprotroph (saprotroph), and endophyte-litter saprotroph-soil saprotroph-undefined saprotroph (saprotroph-symbiotroph) showed significantly lower abundances in Dipsocoromorpha (Fig. S21 in Supplemental File 1). Ericoid mycorrhizal (pathotroph-symbiotroph), undefined saprotroph, and dung saprotroph-wood saprotroph (saprotroph) had significantly lower abundances in Leptopodomorpha. In both semiaquatic Gerromorpha and aquatic Nepomorpha, the abundance of plant pathogens (pathotroph) was significantly lower than that in the remaining samples.

## DISCUSSION

A variety of symbiotic microorganisms colonize the exoskeleton, gut, hemocoel, and even the cells of insects (30). The microbial compositions are affected by various factors, and simultaneously they provide multiple benefits for host insects (14–20), among which the

true bugs are no exception (28–33, 35–39). To investigate the composition patterns of symbiotic microorganisms in the enigmatically diverse true bugs, and their roles in host adaptations to diverse habitats, we surveyed the symbiotic bacteria and fungi of 204 heteropteran samples. These insects represent almost all known habitats and all seven infraorders of true bugs, especially the rare, sampled aquatic and semiaquatic bugs, dipsocoromorphans, and enicocephalomorphans. Except for the bacterial diversity of the superfamily Gerroidea in Gerromorpha, and the bacterial and fungal diversity of Cimicomorpha and Pentatomomorpha, the bacterial and fungal communities of the remaining true bugs studied here have never been included in microbial studies. In the technical aspect, both full-length bacterial 16S and fungal ITS amplicons were sequenced by the PacBio platform, which could minimize the overestimation or underestimation of community diversity (40, 41).

Different compositions of bacterial and fungal communities were detected in true bugs. The symbiotic bacterial compositions of Pentatomomorpha, Cimicomorpha, and the superfamily Gerroidea in Gerromorpha, are consistent with previous studies (35, 37, 38, 42), among which *Wolbachia* and *Rickettsia* are common endosymbionts and can provide nutritional benefits to bugs (35, 38, 43, 44). Meanwhile, the genus *Caballeronia* can help squash bugs prevent the establishment or proliferation of pathogens (45). In contrast, the symbiotic bacteria of aquatic bugs covered the most bacterial phyla but without a dominant, universal, or stable phylum, followed by semiaquatic bugs (Table 3). While, some bacterial phyla, i.e., *Deferribacterota*, *Desulfobacterota*, and *Bacteroidota* (Fig. 1), and orders, i.e., *Oscillospirales*, *Erysipelotrichales*, and *Rhodospirillales* (Fig. S1 in Supplemental File 1) were unique for aquatic bugs, or unique for both aquatic and semiaquatic bugs, which are all closely related to aquatic habitats. Species belonging to the phylum *Deferribacterota* and order *Erysipelotrichales* can also be found in yellow mealworms (46) and moths (47), respectively. Species in the family *Oscillospiraceae* (*Oscillospirales*) can yield the short-chain fatty acid butyrate (48), which was demonstrated to free honeybees from pesticides and pathogen infections (49).

The results of alpha diversity, beta diversity, and statistical analyses showed that the bacterial communities were affected by both the hosts and environmental factors (Fig. 2; Fig. S2 to S7 in Supplemental File 1; Tables S3 to S5 in Supplemental File 1), which is consistent with the results presented in beetles, moths, ants, and brown planthoppers (23, 50–52). In this study, the habitats had a great impact on the bacterial compositions of true bugs, and the symbiotic microbial communities of vegetation-inhabiting and aquatic-dwelling bugs seemed to form two extremes. The terrestrial bugs usually found in vegetation (Cimicomorpha and Pentatomomorpha) showed similar bacterial diversities (Fig. 1 to 3). Their symbiotic bacterial communities were significantly different from those of aquatic bugs.

For terrestrial pentatomomorphans, the interactions between symbiotic bacterial and fungal communities were the simplest, and the interactions within bacterial communities were simple, although not the simplest (Fig. 4; Table 3; Fig. S12 in Supplemental File 1). This simple association may be related to the long coevolution between Pentatomomorpha and bacteria (53, 54). In addition, these plant-feeding true bugs may benefit from bacterial communities in the biodegradation of various xenobiotics, i.e., the components of plant tissues and pesticides (55–58). In Cimicomorpha, the interactions within symbiotic fungal communities were the simplest. Whereas, Cimicomorpha had a distinct fungal genus from other samples (Fig. S10 in Supplemental File 1). Such symbiotic fungi may also have close evolutionary or molecular physiological relationships and provide benefits for their hosts. The results of alpha and beta diversities, heatmap analysis, LEfSe analysis, and co-occurrence networks showed various degrees of relevance between fungal communities and hosts (Fig. 2 to 4; Fig. S9 and S10 in Supplemental File 1), which is difficult to show compared with that of bacterial communities. Nevertheless, fungi have been suggested to provide benefits for insects, as previously reported in beetles and adelgids (59, 60).

For the aquatic nepomorphans, their microbial communities had neither dominant bacteria nor dominant fungi or a universal microbiome present in all samples (Fig. 1 and 3; Fig. S1 and S9 in Supplemental File 1). However, bacteria belonging to multiple phyla

participated in the association according to the networks (Fig. 4). These coordinated bacteria with low abundances may perform the same functions as those of the symbiotic bacterial communities in fruit fly larvae (15). According to the results of bacterial functional prediction, the symbiotic bacterial communities may supplement or collaborate with the host in amino acid and lipid metabolism pathways (Fig. 5). Hypoxia is the main challenge confronting aquatic animals and results in a variety of challenges, including making insects more susceptible to microbial and metazoan infections; upregulating the proline, leucine, asparagine, glutamine, and serine; decreasing the abundance of tyrosine, tryptophan, and methionine; and suppressing the synthesis of fatty acids (61, 62). Hypoxia-induced metabolic perturbations and immune pressures are likely to be relieved by symbiotic bacterial communities. In addition, the prediction of fungal functional guilds showed significantly lower abundances of plant pathogen (pathotroph) in Nepomorpha and Gerromorpha. Insects in these two infraorders were (semi)aquatic bugs and predators. They would rarely be exposed to or need to defend plants as terrestrial true bugs do. In semiaquatic gerromorphans, the diversity of symbiotic microorganisms was similar to that of aquatic nepomorphans, while the interactions within and between bacterial and fungal communities were simpler. The similarities and differences in the symbiotic microorganisms between these two infraorders may be caused by the degrees of correlation with the aquatic habitat, which is reflected in the different degrees of selection pressure faced by these true bugs.

True bugs inhabiting the transition zones between terrestrial and aquatic habitats often had more flexible symbiotic microbial communities. In Dipsocoromorpha and Leptopodomorpha, the networks within and between bacterial and fungal communities showed the most complicated interactions (Fig. 4; Table 3; Fig. S12 in Supplemental File 1). Except for the highly specialized infraorder Nepomorpha, Dipsocoromorpha and Leptopodomorpha are the basal groups of the clades to which they belong (Fig. 3). Their complex networks may indicate the microbial pattern of the common ancestor of the true bugs. Species of Dipsocoromorpha and Leptopodomorpha have great potential in the transition to aquatic or arid terrestrial habitats. From the point of host evolution, the symbiotic microbial co-occurrence networks changed from the most complex types (basal clades) to the simpler or specific types (terminal clades) (Fig. 4). From the perspective of ecology, the diversity of symbiotic microbial communities seems to change gradually, i.e., from true bugs inhabiting terrestrial vegetation to species inhabiting moist areas and semiaquatic habitats to those inhabiting aquatic habitats (Fig. 2). For the true bugs inhabiting the two extreme habitats, symbiotic bacterial communities are likely to help hosts face hypoxia and plant defense (Fig. 5).

Although symbiotic microorganism features of true bugs varied with habitat, the fungal-bacterial co-occurrences were robust and consistent in different infraorders. This type of cross-kingdom network has been revealed in the case of environmental and plant samples (63, 64). In addition, fungal communities can promote bacterial anaerobic metabolism in coastal sediments (65) and affect bacteria in deadwood (66). They may function as a group and even act upstream of the pathways on hosts. Therefore, investigations of both symbiotic bacteria and fungi as well as their association patterns are essential to explore the full-scale aspects of symbiotic microorganisms.

In this study, we explored the diversities and potential functions of symbiotic bacterial and fungal communities during host colonization of aquatic and semiaquatic habitats based on comprehensive taxon sampling. Our results show that symbiotic microorganisms are likely to help true bugs survive diverse selection pressures from their environments and play important roles in the fitness of their insect hosts in diverse habitats. This study lays a solid foundation for further research on the functions and transmission pathways of symbionts in true bugs using metagenomic and/or experimental analyses.

## MATERIALS AND METHODS

**Sample collecting, DNA extraction, and sequencing.** A total of 204 samples were explored, among which 139 belonging to five infraorders were newly collected and sequenced (Table 1). Raw data of the amplicons of the remaining 65 samples belonging to the other two terrestrial infraorders, which were

collected and sequenced by our team as well, were downloaded from the China National Center for Bioinformation (CNCB) with the BioProject number PRJCA007913. Overall, 204 adult heteropteran samples were collected from December 2016 to August 2021 in 14 Chinese provinces and one state of Brazil (Table 1; Table S1 in Supplemental File 1). Considering that the body size of each sample varied from approximately 1 mm to 6 cm and the representativeness of the host species and their corresponding symbiotic microorganisms, each sample contained 1 to 150 male and female individuals. These samples were preserved and processed following the protocols described previously by Yang et al. (41). Briefly, samples were preserved in 100% ethanol at 4°C in the field and preserved in a −80°C freezer immediately after returning to the laboratory. Before DNA extraction, samples were surface sterilized with 70% ethanol for at least 5 min and then rinsed with phosphate-buffered saline (PBS) three times. Next, the wings and legs for each sample were removed with tweezers, and the remnants were cut into pieces. DNA was extracted with the MoBio Powersoil kit (Qiagen, Hilden, Germany) following the instructions of the manufacturer. The full-length 16S rRNA gene of bacteria and ITS region of fungi were amplified and sequenced on the PacBio Sequel II platform with the primer sets listed in Table S6 in Supplemental File 1 (67, 68).

**Analyses of microbial community and diversity.** The 204 samples were classified into different groups according to China's eco-geographical region map and Heihe-Tengchong Line (69). Circular consensus sequencing (CCS) reads for each sample were quality controlled using the DADA2 package v1.22.0 in R v4.1.2 (70). Then, the feature table and sequences were imported into QIIME2 pipeline v2021.11 (71) to perform further analyses. The SILVA database v138.1 for the 16S rRNA gene and the UNITE database v8.2 (https://docs.qiime2.org/2022.2/data-resources/) for the ITS region were used in taxonomic classification. The bacterial ASVs (amplicon sequence variants) assigned to mitochondrial and chloroplast sequences and those with less than 50 counts were removed from the data sets. Fungal ASVs with less than 30 counts were filtered out. Rarefaction curves were generated by the QIIME2 pipeline, and all of them reached a plateau when the sequencing depths were over 200 (Fig. S22 in Supplemental File 1). The bacterial and fungal data sets output from the QIIME2 pipeline can be found on DataOpen (http://www.dataopen.info/home/accountdatafile/add/id/262).

The alpha diversity indices, including the Shannon diversity index, observed features, Faith's phylogenetic diversity (Faith's PD), and Pielou's evenness, were calculated by the "q2-diversity" plugin of QIIME2 pipeline v2021.11 (71). To calculate the pairwise differences among groups, phylogenetic ANOVA analysis, LSD *post hoc* test, and Kruskal-Wallis test were performed by R package phytools v1.0 (72), QIIME2 pipeline v2021.11 (71), and SPSS v20.0.0 (73), respectively. The $P$ values for pairwise comparisons were adjusted with the FDR (false discovery rate) method.

In the analysis of beta diversity, unweighted and weighted UniFrac distances were calculated by the "q2-diversity" plugin of the QIIME2 pipeline. The R packages ape v5.6 (74), vegan v2.5.7 (75), and ggplot2 v3.3.5 (76) were used to generate the PCoA plot based on the unweighted and weighted UniFrac distance matrices. PERMANOVA test was performed to calculate the pairwise differences among groups. The $P$ values for pairwise comparisons were adjusted with the FDR method.

To survey the relationships between the genetic distance of the host and the unweighted UniFrac distance of microbial communities and the relationships between the geographical distance of the host and the unweighted UniFrac distance of microbial communities, the R packages vegan v2.5.7 (75), geosphere v4.1.3 (https://github.com/rspatial/geosphere), and ggplot2 v3.3.5 (76) were used to perform Mantel and Procrustes tests. Among the overlapping bacterial and fungal samples, Mantel and Procrustes tests were also performed to evaluate the correlations between bacterial and fungal unweighted UniFrac distances. The heatmap was generated by TBtools (77).

**LEfSe and co-occurrence network analyses.** Linear discriminant analysis effect size (LEfSe) analysis was performed at the genus level on the website http://huttenhower.sph.harvard.edu/galaxy/root/index (78) with an alpha value threshold of 0.01.

To generate microbial community co-occurrence networks, the R package Hmisc v4.7-0 (https://hbiostat.org/R/Hmisc/) was used to perform correlation analysis based on genus-level abundances. The top 120 most abundant bacterial ASVs and the top 120 most abundant fungal ASVs were selected. Correlation coefficients greater than 0.8 or less than −0.8 with a corresponding $P$ value less than 0.001 were included to generate the networks. The network structure was visualized with Gephi v0.9.5 (79).

**Functional prediction.** The functions of bacterial communities were predicted by PICRUSt2 v2.4.2 (80). The fungal ecological guilds were inferred using FUNGuild v1.0 (81). Guild assignments with "probable" and "highly probable" confidences were accepted. The results of functions and abundances were analyzed with STAMP v2.1.3 (82) to calculate the significantly different functions between different groupings. For each infraorder, Welch's $t$ tests (two-sided) were used for the comparison with all other samples. The $P$ value threshold was set as 0.001 for bacterial data and 0.01 for fungal data.

**Phylogenetic relationships of host species.** To reconstruct the phylogenetic relationships of host species, the legs of samples were used to exact total genomic DNA using the TIANamp Micro DNA kit (TIANGEM, Beijing, China) according to the instructions of the manufacturer. The mitochondrial genes COI and COII and nuclear genes 18S and 28S rRNA genes were amplified in one to six overlapping fragments. PCR products were sequenced by the Sanger method, and the results were assembled by SeqMan in the DNAStar v7.1.0 program package (83). The primer sets used for amplification and sequencing are listed in Table S6 in Supplemental File 1. Thirteen species belonging to the other three suborders of Hemiptera were selected as outgroups. The corresponding sequences were downloaded from the NCBI database (www.ncbi.nlm.nih.gov) (Table S7 in Supplemental File 1).

Nucleotide sequences of the two mitochondrial genes were converted into amino acid sequences and then individually aligned in MEGA v7 (84). The aligned amino acid sequences were converted back to nucleotide sequences. The 1st and 2nd codon positions of nucleotides were used for further

phylogenetic analyses. For the two nuclear genes, sequences were individually aligned with SeaView v4 (85) and then manually optimized according to the secondary structures of 18S and 28S rRNAs (86). Then, all four aligned nucleotide sequences were concatenated using Sequence Matrix v1.7.8 (87). The concatenated matrix can be found on DataOpen (http://www.dataopen.info/home/accountdatafile/add/id/262). The concatenated matrix consisted of 7 778 nucleotide positions. The statistical information for the concatenated matrix for each species is shown in Table S8 in Supplemental File 1. The overall coverage of the four genes was 92.5%, and that of the nucleotide sites was 79.4%. The best substitution models were calculated using IQ-TREE v1.6.12 (88). Phylogenetic analysis was performed with RAxML v8.2.8 (89) using a rapid bootstrap algorithm with 1 000 replicates. The clades Auchenorrhyncha and Tripartita (Ochteroidea + Notonectoidea + Naucoroidea) were constrained according to independent evidence from the morphology, transcriptome, and mitogenome (8, 90–92). Bootstrap values were calculated using BOOSTER (93) with default settings.

**Data availability.** The R scripts and data sets output from the QIIME2 pipeline are available on DataOpen (http://www.dataopen.info/home/accountdatafile/add/id/262). The newly generated sequences of hosts (accession numbers OP393935 to OP394045, OP410481 to OP410736, OP435260 to OP435267, and OP435348; project code SSLL) and the raw data of amplicon of symbionts (BioProject accession number PRJNA875249) are available in NCBI database and the Bold systems (Barcode Of Life Data System).

## SUPPLEMENTAL MATERIAL

Supplemental material is available online only.
**SUPPLEMENTAL FILE 1**, PDF file, 6.1 MB.

## ACKNOWLEDGMENTS

We sincerely thank Zu-Qi Mai (School of Life Sciences, Sun Yat-sen University, Guangzhou, China) for his kind providing of *Lethocerus indicus* images in Fig. 1.

This work was supported by the National Natural Science Foundation of China (grant number 31222051). FFFM was funded by the Fundação Carlos Chagas Filho de Amparo à Pesquisa do Estado do Rio de Janeiro (FAPERJ; grant numbers E-26/201.362/2021 and E-26/203.250/2021) and the Conselho Nacional de Desenvolvimento Científico e Tecnológico (CNPq; grant number 301942/2019-6).

Q.X. and Y.-h.W. conceived and designed the project. Q.X., Y.-h.W., Z.-w.Y., and Y.M. designed the experiments. Y.M., Z.-w.Y., J.-y.L., P.-p.C., F.F.F.M., Z.-h.L., J.-d.Y., B.-j.X., Y.-h.W., and Q.X. collected the true bug samples of this study. J.-y.L., P.-p.C., and Y.-h.W. identified the true bug samples. Y.M., Z.-w.Y., and Z.-h.L. performed the experiments. Y.M., Z.-w.Y., Y.-h.W., and Q.X. analyzed the data. Y.M. wrote the manuscript. Z.-w.Y., P.-p.C., F.F.F.M., Y.-h.W., and Q.X. revised the manuscript. All the authors have read and approved the submission for publication.

We declare no competing interests.

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
