## [Reviewer comments · Microbiology Spectrum]

Microbiology Spectrum

Symbiotic microorganisms and their different association types in aquatic and semi-aquatic bugs

Yu Men, Zi-wen Yang, Jiu-yang Luo, Ping-ping Chen, Felipe Ferraz Figueiredo Moreira, Zhi-Hui Liu, Jia-dong Yin, Bao-jun Xie, Yan-hui Wang, and Qiang Xie

Corresponding Author(s): Yan-hui Wang, Sun Yat-sen University

Review Timeline:

Submission Date:	July 20, 2022
Editorial Decision:	August 17, 2022
Revision Received:	September 17, 2022
Editorial Decision:	October 17, 2022
Revision Received:	October 27, 2022
Accepted:	November 3, 2022

Editor: John Chaston

Reviewer(s): Disclosure of reviewer identity is with reference to reviewer comments included in decision letter(s). The following individuals involved in review of your submission have agreed to reveal their identity: John G McMullen (Reviewer #1)

Transaction Report:

DOI: <https://doi.org/10.1128/spectrum.02794-22>

August 16, 2022

Dr. Yan-hui Wang
Sun Yat-sen University
State Key Laboratory of Biocontrol
No. 135 West Xingang Road
Guangzhou, Guangdong
China

Re: Spectrum02794-22 (**Symbiotic microorganisms and their different coordination types in aquatic and semi-aquatic bugs**)

Dear Dr. Yan-hui Wang:

Thank you for submitting your manuscript to Microbiology Spectrum. It has been reviewed by two experts, and both viewed the manuscript positively and had some suggestions for revision. I have therefore decided to request 'Modifications', which means that if you choose to, you can submit a revised version of the text following the details outlined below, and that it is likely that I will send the revision out for a second round of review.

You will see that Reviewer 1 has suggested several additional analyses, and I generally agree with Reviewer 1's suggestions, especially where the analysis is critical or fairly straightforward (for example, analyzing additional measures of alpha or beta diversity). As you revise, I confirm to you that Spectrum's scope is that "rather than making subjective evaluations of potential impact, Microbiology Spectrum publishes research studies that are of high technical quality and are useful to the community." I will be supportive of revisions that meet this expectation.

Link Not Available

Sincerely,

John Chaston

Journals Department
Reviewer comments:

Reviewer #1 (Comments for the Author):

Symbiotic microorganisms and their different coordination types in aquatic and semi-aquatic bugs

Summary: There is a dearth of information regarding profiling of microorganisms associated with aquatic and semi-aquatic insects. The authors of this study surveyed approx. 200 insect species across the suborder Heteroptera and their various habitats. The authors surveyed the microbiome using full length 16S and ITS genes using PacBio. Bacterial diversity varied by host habitat, while fungal communities differed by host geographic location. Further functional prediction data indicates potential ways that the microbiome may benefit their host across the diverse habitats.

What an impressive feat! Congrats to the authors for collecting so many insects in so many locations.

Major:

-This is a take or leave it comment, but the Introduction could be restructured a bit in my opinion. The first paragraph feels like a long list, while the second paragraph has a lot of nice opening material about the aquatic lifestyle. It could be nice to flip these to highlight why the Heteroptera are a great group of insects to study and compare to terrestrial environments, which have been extensively studied for terrestrial insects. We could then hypothesize that given the stark differences between these two habitats, microbial phenotypes that may benefit their hosts could vary from terrestrial systems.

-For alpha diversity, it would be nice to have the full set of analyses: richness, diversity (e.g., Shannon), and evenness alongside the phylogenetic diversity index. I think these could be displayed as a table.

-For the ANOVA type analyses (Fig 2a,b,d,e), it is critical to also do a pglS (phylogenetic generalized least squares) or a phylogenetic ANOVA analysis to take host phylogeny into account. This is a good way to take into account host phylogeny given the non-independence of the data.

-For the overlap in sampling in bacteria and fungi, it could be good to also do a Procrustes analysis to see if the two ordinations are correlated. They seem like they have different processes operating on their assemblages, and this could be a nice way to show their level of association with one another early on in the manuscript.

-Final paragraph in the Discussion: It is important to keep the language as suggestive. None of the results directly related to causation, but are only correlative. Some of the microbes may benefit their hosts, some may not, and some could also be transient microbes with little effects on their hosts.

-In the discussion, it would be good to touch on the fungal-bacterial cooccurrences, especially if there are any notable interactions since this is an emerging theme in microbiome sciences.

-There are many taxa studied in this paper, but it could be nice to highlight some of the microbes in the Discussion (especially some of the easy ones like endosymbionts, which have such a strong track record for their presence among hemipterans).

-In the supplement, please include if insect sex is known.

-It would be great to complement the bacterial PICRUST data with FUNGuild to show the ecological guilds among the fungi found.

-How would a weighted UniFrac PCoA look? It could be good to add this into the supplement.

-For figures with the infraorders color coded, the two green colors look the same, please change to something more distinguishing

-Fig 2, please indicate how the phylogenetic diversity was statistically analyzed in the text. It seems unclear as is. It could be useful to also do some posthoc tests as well to compare among taxa groups.

-L394: Make sure to adjust the pvalue for pairwise comparisons with FDR or similar method.

-All the sequence data needs to be deposited to the correct repository.

-For R scripts, it would be great if those were available to help make the science reproducible.

-It would be good to have your dataset output from QIIME included in the supplement as well as a rarefaction curve for sequencing.

Minor:

Overall the manuscript reads well, but there are spelling/grammatical mistakes throughout that need to be fixed.

L34: maybe use a different phrase for coordination type to be clearer.

L110: In total instead of totally

L123/135: I'd suggest changing features to ASVs so that readers don't have to go the Methods to check threshold for species classifications.

L201-203: It could be more helpful to describe this a bit, otherwise it is a bit unclear what additional information this figure is providing. It seems really interesting from looking at it (for instance this figure has a high abundance of endosymbionts).

L242: It could also be that the microbes are occupying similar niches in the insect (especially if they are more closely, phylogenetically related taxa) and that is also why we see positive associations as well as potential cooperation.

Fig 1: It could be nice to indicate habitats in this figure since there are many insect taxa.

Table S3: Are these all p-values? Please indicate what information is in the table.

Table 2: Please indicate what the / is. Also change more regulation to something more specific to the dataset (enriched perhaps?)

Reviewer #2 (Comments for the Author):

General comments: This study describes bacterial and fungal communities in hemipteran insects inhabiting the terrestrial, semi-aquatic and aquatic environments. The methods used are appropriate. The paper is well-written and touches on interesting points, such as fungal and bacterial community associated with semi-aquatic and aquatic insects, and co-occurrence networks that showed the bugs inhabiting similar habitats shared similar microbial types. The result of this study can serve as reference for future studies investigating the microbial communities associated with insects inhabiting different ecological niches.

Abstract:

Lines 34-37 - Functional pathway abundances of the bacterial community was predicted based on the identity of the bacterial communities. These lines could be rephrased to indicate that the functions are based on predictive analysis.

Introduction:

It is well thought out and comprehensive

Results:

- One of the advantages of using Pacbio sequencing is the ability to obtain full length 16s reads and ITS reads that helps with better taxonomic resolution. Adding species level information on bacterial and fungal communities associated with the insects could provide valuable insights to the wider scientific community.

- Line 245 - "Functions of symbiotic bacterial communities" can be rephrased to indicate its hypothetical/predictive

Discussion:

- Is well composed and covers major points of the study

- Hemipteran insect order especially the terrestrial bugs is one of the well-studied in terms of insect symbiosis interaction. The Discussion can be improved with the addition of few lines to establish if this study was able to identify the well documented bacterial symbionts of the terrestrial bugs belonging to Pentatomomorpha and Cimicomorpha.

Materials and Methods:

- In Sample collecting, DNA extractions and sequencing section, please add information about the primers that were used to amplify the 16s and ITS region.

Figure2: In the plots a and d - The box plots could be colored to indicate the infraorder than the x-axis text. This would be help in understanding the figures easier.

Staff Comments:

Preparing Revision Guidelines

Please return the manuscript within 60 days; if you cannot complete the modification within this time period, please contact me. If you do not wish to modify the manuscript and prefer to submit it to another journal, please notify me of your decision immediately so that the manuscript may be formally withdrawn from consideration by Microbiology Spectrum.

Reviewer #1 (Comments for the Author):

Summary: There is a dearth of information regarding profiling of microorganisms associated with aquatic and semi-aquatic insects. The authors of this study surveyed approx.. 200 insect species across the suborder Heteroptera and their various habitats. The authors surveyed the microbiome using full length 16S and ITS genes using PacBio. Bacterial diversity varied by host habitat, while fungal communities differed by host geographic location. Further functional prediction data indicates potential ways that the microbiome may benefit their host across the diverse habitats. What an impressive feat! Congrats to the authors for collecting so many insects in so many locations.

Answer: Thank you very much for your positive and helpful comments. We have supplemented a series of analyses regarding profiling of microorganisms associated with aquatic and semi-aquatic insects in the revised manuscript. Below, please find our point-by-point responses to your comments.

Major:

Question #1

-This is a take or leave it comment, but the Introduction could be restructured a bit in my opinion. The first paragraph feels like a long list, while the second paragraph has a lot of nice opening material about the aquatic lifestyle. It could be nice to flip these to highlight why the Heteroptera are a great group of insects to study and compare to terrestrial environments, which have been extensively studied for terrestrial insects. We could then hypothesize that given the stark differences between these two habitats, microbial phenotypes that may benefit their hosts could vary from terrestrial systems.

Answer: We accept this suggestion and have adjusted the order of the first two paragraphs in the introduction section.

Question #2

-For alpha diversity, it would be nice to have the full set of analyses: richness, diversity (e.g., Shannon), and evenness alongside the phylogenetic diversity index. I think these could be displayed as a table.

Answer: We accept this recommendation and have added the full set of alpha diversity analyses, including observed features, piou's evenness, and shannon entropy index. The results can be found in the Table 2. At the same time, the results of alpha diversity analyses are also shown in box-plots in Supplementary Figs. S2, S3, S6, and S7.

Question #3

-For the ANOVA type analyses (Fig 2a,b,d,e), it is critical to also do a pglS (phylogenetic generalized least squares) or a phylogenetic ANOVA analysis to take host phylogeny into account. This is a good way to take into account host phylogeny given the non-independence of the data.

Answer: We accept this comment and have performed a phylogenetic ANOVA analysis to evaluate the differences of alpha diversity indexes between groups. The results have been added to the "Factors influencing bacterial and fungal communities" section and the Supplementary Tables S3 and S5.

Question #4

-For the overlap in sampling in bacteria and fungi, it could be good to also do a Procrustes analysis to see if the two ordinations are correlated. They seem like they have different processes operating on their assemblages, and this could be a nice way to show their level of association with one another early on in the manuscript.

Answer: We agree with this comment. We have performed the Procrustes and Mantel tests to evaluate correlations between the two ordinations. The corresponding descriptions have been added to the Results section as follows. “In order to explore the relationships between symbiotic bacterial and fungal communities, Mantel test and Procrustes analysis were performed based on unweighted UniFrac distances. Results of Mantel test and Procrustes analysis showed significant correlations ($p = 0.039$) and no statistic relevance ($p = 0.079$) between bacterial and fungal communities, respectively (Supplementary Fig. S11). The difference reflected different level of associations among various symbionts.”

Question #5

-Final paragraph in the Discussion: It is important to keep the language as suggestive. None of the results directly related to causation, but are only correlative. Some of the microbes may benefit their hosts, some may not, and some could also be transient microbes with little effects on their hosts.

Answer: We accept this comment and have revised the final paragraph of Discussion. The revised paragraph is as follows: “In this study, we explored the diversities and potential functions of symbiotic bacterial and fungal communities during the host colonizing aquatic and semi-aquatic habitats based on a comprehensive taxon sampling. Our results show that symbiotic microorganisms are likely to help true bugs survive diverse selection pressures from their environments and play important role for the fitness of their insect host to diverse habitats. This study lays a solid found for further research on the functions and transmission pathways of symbionts in true bugs using metagenomic and/or with experimental analyses.”

Question #6

-In the discussion, it would be good to touch on the fungal-bacterial cooccurrences, especially if there are any notable interactions since this is an emerging theme in microbiome sciences.

Answer: Thank you for your good suggestion and we have added a paragraph to discuss the fungal-bacterial co-occurrences (see the last but one paragraph of Discussion section). The corresponding text is as follows: “Although symbiotic microorganism features of true bugs varied with habitats, the fungal-bacterial co-occurrences were robust and consistent in different infraorders. This kind of cross-kingdom networks have been revealed in the case of environment and plant samples (63; 64). Besides, the fungal communities can promote bacterial anaerobic metabolisms in coastal sediments (65) and affect bacteria in deadwood (66). They may play functions as a group and act as the upstream of the pathways affected on hosts as well. Therefore, investigations of both symbiotic bacteria and fungi as well as their association pattern were essential to explore full-scale aspects of symbiotic microorganisms.”

Question #7

-There are many taxa studied in this paper, but it could be nice to highlight some of the microbes in the Discussion (especially some of the easy ones like endosymbionts, which have such a strong track record for their presence among hemipterans).

Answer: We accept this recommendation and have highlighted the significantly abundant symbionts in terrestrial true bugs and the unique symbionts in (semi-)aquatic bugs. We also presented the proved functions of these symbionts in other species. The corresponding content can be found in the second paragraph of Discussion section.

Question #8

-In the supplement, please include if insect sex is known.

Answer: I would like to give some explanations on this question. "Considering the body size of each species varying from about 1 mm to 6 cm and the representativeness of host species and their corresponding symbiotic microorganisms, each sample contained one to 150 male and female individuals." Almost all samples contain both male and female specimens.

Question #9

-It would be great to complement the bacterial PICRUST data with FUNGuild to show the ecological guilds among the fungi found.

Answer: Thank you for this helpful recommendation. We agree with this view and have inferred the fungal guilds using FUNGuild. The related methods, results, and discussions have been added to the revised manuscript.

Question #10

-How would a weighted UniFrac PCoA look? It could be good to add this into the supplement.

Answer: We accept this comment and have added the weighted UniFrac PCoA plots into the Supplementary Fig. S4. The results is similar to that of unweighted UniFrac PCoA plots.

Question #11

-For figures with the infraorders color coded, the two green colors look the same, please change to something more distinguishing

Answer: We agree and accept this recommendation. We have changed the color of Cimicomorpha into light green in all figures and supplementary figures.

Question #12

-Fig 2, please indicate how the phylogenetic diversity was statistically analyzed in the text. It seems unclear as is. It could be useful to also do some post hoc tests as well to compare among taxa groups.

Answer: We accept this suggestion. We have added the methods of calculating and statistic analyzing the alpha diversity indexes in the method section. At the same time, LSD-Post hoc test was performed to compare among the seven infraorders. The methods and results of LSD-Post hoc test were added to the corresponding parts in the revised manuscript.

Question #13

-L394: Make sure to adjust the p value for pairwise comparisons with FDR or similar method.

Answer: Thank you very much to remind us of the adjusted p value. We have calculated the FDR adjusted p value for all pairwise comparisons. The corresponding results were also revised.

Question #14

-All the sequence data needs to be deposited to the correct repository.

Answer: Sure, we have submitted all of the raw data to public databases. “The newly generated sequences of hosts (Accession numbers: OP393935–OP394045, OP410481–OP410736, OP435260–OP435267, OP435348; Project code: SSSL) and the raw data of amplicon of symbionts (BioProject number: PRJNA875249) are available in NCBI database and the Bold systems (Barcode Of Life Data System).”

Question #15

-For R scripts, it would be great if those were available to help make the science reproducible.

Answer: We accept this suggestion. The R scripts performed for (1) quality control by Dada2, (2) phylogenetic ANOVA analysis by phytools, (3) PCoA plot by ape and vegan, (4) mantel and procrustes tests by vegan and geosphere, and (5) co-occurrence network by Hmisc have been packaged and submitted to DataOpen (<http://www.dataopen.info/home/accountdatafile/add/id/262>).

Question #16

-It would be good to have your dataset output from QIIME included in the supplement as well as a rarefaction curve for sequencing.

Answer: We accept this comment and the bacterial and fungal datasets output from QIIME2 pipeline (including the results of alpha diversity, beta diversity and rarefaction curve) can be found on DataOpen (<http://www.dataopen.info/home/accountdatafile/add/id/262>). Besides, rarefaction curves of bacterial and fungal samples for different infraorders were uploaded as Supplementary Fig. S22.

Minor:

Question #17

Overall the manuscript reads well, but there are spelling/grammatical mistakes throughout that need to be fixed.

Answer: Thank you for your meticulous review. We have carefully checked the spelling and syntax errors throughout the manuscript.

Question #18

L34: maybe use a different phrase for coordination type to be clearer.

Answer: We accept this suggestion and have replaced the “coordination types” with “association types”.

Question #19

L110: In total instead of totally

Answer: We accept this suggestion and have replaced the “Totally” with “In total”.

Question #20

L123/135: I'd suggest changing features to ASVs so that readers don't have to go the Methods to check threshold for species classifications.

Answer: We accept this suggestion and the “features” have been revised to “ASVs”.

Question #21

L201-203: It could be more helpful to describe this a bit, otherwise it is a bit unclear what additional information this figure is providing. It seems really interesting from looking at it (for instance this figure has a high abundance of endosymbionts).

Answer: We accept this comment and have described the heat map of symbiotic microbial communities at genus level in detail. And the results of LEfSe analysis are looking more clearly in the revised manuscript.

Question #22

L242: It could also be that the microbes are occupying similar niches in the insect (especially if they are more closely, phylogenetically related taxa) and that is also why we see positive associations as well as potential cooperation.

Answer: Thank you for your enlightening comment and providing a novel possibility of symbionts associations. We have revised the corresponding text as follows. “Noting that in the network of Dipsocoromorpha, a group of bacterial ASVs showed negative correlations with a group of fungal ASVs. In the group of bacterial ASVs, 77% of them belonged to family *Budviciaceae* (*Proteobacteria: Gammaproteobacteria*). The two negatively correlated groups of bacterial and fungal symbionts may occupy similar niches in hosts. In addition to this case, almost all ASVs in the remaining co-occurrence networks were positively correlated, suggesting the potential cooperation as well as positive associations among bacterial and fungal communities of true bugs.”

Question #23

Fig 1: It could be nice to indicate habitats in this figure since there are many insect taxa.

Answer: We agree with this comment and have added habitats in revised Fig. 1.

Question #24

Table S3: Are these all p values? Please indicate what information is in the table.

Answer: Table S3 has been renamed as Table S4 in the revised manuscript. It shows the statistical analyses (PERMANOVA tests) for the results of beta diversity based on weighted and unweighted UniFrac distances. These are *p* values and adjusted *p* values.

Question #25

Table 2: Please indicate what the / is. Also change more regulation to something more specific to the dataset (enriched perhaps?)

Answer: The “/” indicate the corresponding analyses were not applicable due to insufficient

samples (<4 samples). We have changed “Higher” and “Lower” to “Enriched” and “Limited” respectively in the revised Table 3.

Reviewer #2 (Comments for the Author):

General comments: This study describes bacterial and fungal communities in hemipteran insects inhabiting the terrestrial, semi-aquatic and aquatic environments. The methods used are appropriate. The paper is well-written and touches on interesting points, such as fungal and bacterial community associated with semi-aquatic and aquatic insects, and co-occurrence networks that showed the bugs inhabiting similar habitats shared similar microbial types. The result of this study can serve as reference for future studies investigating the microbial communities associated with insects inhabiting different ecological niches.

Answer: Thank you for your constructive and kindly comments. As shown below, we have accepted all of your recommendations and revised the manuscript carefully.

Abstract:

Question #1

Lines 34-37 - Functional pathway abundances of the bacterial community was predicted based on the identity of the bacterial communities. These lines could be rephrased to indicate that the functions are based on predictive analysis.

Answer: We accept this suggestion and have revised the corresponding sentence as “Moreover, functional prediction analyses showed that symbiotic bacterial community of aquatic species displayed richer amino acid and lipid metabolism pathways, while plant-feeding true bugs were more benefited from symbiont-provided xenobiotics biodegradation pathway.”

Introduction:

It is well thought out and comprehensive

Response: Thank you for your approval of the Introduction section.

Results:

Question #2

- One of the advantages of using Pacbio sequencing is the ability to obtain full length 16s reads and ITS reads that helps with better taxonomic resolution. Adding species level information on bacterial and fungal communities associated with the insects could provide valuable insights to the wider scientific community.

Answer: Thank you for your recognition and helpful recommendation. We have checked through the manuscript and added species level information on bacterial and fungal communities.

Question #3

- Line 245 - "Functions of symbiotic bacterial communities" can be rephrased to indicate its hypothetical/predictive

Answer: We accept this comment and have rephrased the original sentence to “Functional predictions of symbiotic bacterial and fungal communities”.

Discussion:- Is well composed and covers major points of the study

Response: Thank you for your approval of the Discussion section.

Question #4

- Hemipteran insect order especially the terrestrial bugs is one of the well-studied in terms of insect symbiosis interaction. The Discussion can be improved with the addition of few lines to establish if this study was able to identify the well documented bacterial symbionts of the terrestrial bugs belonging to Pentatomomorpha and Cimicomorpha.

Answer: We accept this comment. We have compared the composition of symbiotic community of terrestrial bugs in this study with that presented in previous works. The corresponding content can be found in the second paragraph of Discussion section.

Materials and Methods:

Question #5

- In Sample collecting, DNA extractions and sequencing section, please add information about the primers that were used to amplify the 16s and ITS region.

Answer: We agree with this comment. The information about the primers that were used to amplify and sequence the two regions of symbiotic microorganisms (16s rRNA gene and ITS region), and the four genes of hosts (COI, COII, 18S and 28S rDNA) can be found in Supplementary Table S6.

Question #6

Figure2: In the plots a and d - The box plots could be colored to indicate the infraorder than the x-axis text. This would be help in understanding the figures easier.

Answer: We accept this comment and have colored the box plots in Figure 2a, d.

October 17, 2022

Dr. Yan-hui Wang
Sun Yat-sen University
State Key Laboratory of Biocontrol
No. 135 West Xingang Road
Guangzhou, Guangdong
China

Re: Spectrum02794-22R1 (**Symbiotic microorganisms and their different association types in aquatic and semi-aquatic bugs**)

Dear Dr. Yan-hui Wang:

Thank you for submitting your manuscript to Microbiology Spectrum. As you will see your paper is very close to acceptance. Please modify the manuscript to adjust the writing in the section as recommended by Reviewer 2. As these revisions are quite minor, I expect that you should be able to turn in the revised paper in less than 30 days, if not sooner. You will find the reviewers' comments below, and I do not intend to send it back out for review - I will be able to evaluate the changes if you add a tracked changes version (instructions below).

When submitting the revised version of your paper, please provide (1) point-by-point responses to the issues raised by the reviewers as file type "Response to Reviewers," not in your cover letter, and (2) a PDF file that indicates the changes from the original submission (by highlighting or underlining the changes) as file type "Marked Up Manuscript - For Review Only". Please use this link to submit your revised manuscript. Detailed instructions on submitting your revised paper are below.

Link Not Available

Sincerely,

John Chaston

Reviewer comments:

Reviewer #1 (Comments for the Author):

Congrats to the authors for addressing all of the reviewer comments. I feel that my previous comments were sufficiently addressed. I think the manuscript reads much better than before (there are still some issues with grammar/syntax, but mostly minor).

I do not have any additional suggestions for the authors.

Reviewer #2 (Comments for the Author):

I appreciate the authors revising the manuscript based on the suggestions from the reviewer's comments. The manuscript is greatly improved from the previous version. I have the following feedback after reviewing the manuscript.

In Results sections under the "Pattern of abundant symbionts according to the phylogeny of host insects" section, the paragraph

focusing on the genus level exploration of the data. The sentences are incomplete and could be improved with rephrasing.

For example - 'An uncultured genus in order Enterobacterales and the genus Rickettsiella (Proteobacteria: Gammaproteobacteria) did not present in Leptopodomorpha, Cimicomorpha, and Pentatomomorpha.' - Instead of 'not present' it should be 'absent'. The sentence doesn't specify the which insect infraorder harbored the bacterial taxa.

I recommend editing this section to focus on genus level bacterial and fungal taxa that were most abundant in the different infraorder. Which genera were most differentially abundant between the terrestrial and aquatic bugs.

Preparing Revision Guidelines

Please return the manuscript within 60 days; if you cannot complete the modification within this time period, please contact me. If you do not wish to modify the manuscript and prefer to submit it to another journal, please notify me of your decision immediately so that the manuscript may be formally withdrawn from consideration by Microbiology Spectrum.

Reviewer #1 (Comments for the Author):

Congrats to the authors for addressing all of the reviewer comments. I feel that my previous comments were sufficiently addressed. I think the manuscript reads much better than before (there are still some issues with grammar/syntax, but mostly minor). I do not have any additional suggestions for the authors.

Answer: Thank you very much for your constructive comments at the first round of review. For your concerns on grammar/syntax, we have carefully checked the manuscript, and purchased a professional language editing service of AJE (American Journal Experts).

Reviewer #2 (Comments for the Author):

I appreciate the authors revising the manuscript based on the suggestions from the reviewer's comments. The manuscript is greatly improved from the previous version. I have the following feedback after reviewing the manuscript.

Answer: Thank you for your positive comments. We have accepted all of your recommendations below and revised the manuscript carefully.

Question #1

In Results sections under the "Pattern of abundant symbionts according to the phylogeny of host insects" section, the paragraph focusing on the genus level exploration of the data. The sentences are incomplete and could be improved with rephrasing.

For example - 'An uncultured genus in order Enterobacteriales and the genus *Rickettsiella* (Proteobacteria: Gammaproteobacteria) did not present in Leptopodomorpha, Cimicomorpha, and Pentatomomorpha.' - Instead of 'not present' it should be 'absent'. The sentence doesn't specify the which insect infraorder harbored the bacterial taxa.

Answer: We have followed this comment and rephrased the sentence as follows. "An uncultured genus in the order *Enterobacteriales* and the genus *Rickettsiella* (*Proteobacteria: Gammaproteobacteria*) were absent in Leptopodomorpha, Cimicomorpha, and Pentatomomorpha. In other words, these bacteria were only present in the remaining infraorders, in which most bugs live in humid and (semi-)aquatic areas."

Question #2

I recommend editing this section to focus on genus level bacterial and fungal taxa that were most abundant in the different infraorder. Which genera were most differentially abundant between the terrestrial and aquatic bugs.

Answer: We agree with this comment and revised the corresponding paragraph as follows. "At the genus level, the losses and acquisitions of symbionts among host superfamilies were also provided (Supplementary Fig. S9). For bacterial communities, the genus *Wolbachia* (*Proteobacteria: Alphaproteobacteria*) was present in all seven

infraorders. An uncultured genus in the order *Enterobacterales* and the genus *Rickettsiella* (*Proteobacteria: Gammaproteobacteria*) were absent in Leptopodomorpha, Cimicomorpha, and Pentatomomorpha. In other words, these bacteria were only present in the remaining infraorders, in which most bugs live in humid and (semi-)aquatic areas. In addition, there was no *Burkholderia* (*Proteobacteria: Gammaproteobacteria*) in aquatic Nepomorpha. The genera *Pantoea*, *Pectobacterium*, and *Caballeronia* (*Proteobacteria: Gammaproteobacteria*) were mainly present in terrestrial bugs. In fungal communities, the genus *Naganishia* (*Basidiomycota: Tremellomycetes*) was absent in terrestrial bugs (Cimicomorpha and Pentatomomorpha). An unidentified genus in the phylum *Chytridiomycota* was absent in terrestrial Cimicomorpha as well.”

November 3, 2022

Dr. Yan-hui Wang
Sun Yat-sen University
State Key Laboratory of Biocontrol
No. 135 West Xingang Road
Guangzhou, Guangdong
China

Re: Spectrum02794-22R2 (**Symbiotic microorganisms and their different association types in aquatic and semi-aquatic bugs**)

Dear Dr. Yan-hui Wang:

Thank you for submitting your revised work. Your manuscript has been accepted, and I am forwarding it to the ASM Journals Department for publication. You will be notified when your proofs are ready to be viewed.

Sincerely,

John Chaston
Editor, Microbiology Spectrum
